# Sequence-Based Viscosity Prediction for Rapid Antibody Engineering

**DOI:** 10.3390/biom14060617

**Published:** 2024-05-23

**Authors:** Bram Estes, Mani Jain, Lei Jia, John Whoriskey, Brian Bennett, Hailing Hsu

**Affiliations:** 1Amgen Research, Protein Therapeutics, Thousand Oaks, CA 91320, USA; mjain07@amgen.com (M.J.); leijiachem@gmail.com (L.J.); 2Amgen Research, Inflammation, Thousand Oaks, CA 91320, USA; jwhorisk@amgen.com (J.W.); bennett@amgen.com (B.B.); hhsu@amgen.com (H.H.)

**Keywords:** therapeutic antibody, mAb, viscosity, machine learning, predictive model, interleukin 13 (IL-13), protein structure, protein engineering, immunoglobulin G (IgG)

## Abstract

Through machine learning, identifying correlations between amino acid sequences of antibodies and their observed characteristics, we developed an internal viscosity prediction model to empower the rapid engineering of therapeutic antibody candidates. For a highly viscous anti-IL-13 monoclonal antibody, we used a structure-based rational design strategy to generate a list of variants that were hypothesized to mitigate viscosity. Our viscosity prediction tool was then used as a screen to cull virtually engineered variants with a probability of high viscosity while advancing those with a probability of low viscosity to production and testing. By combining the rational design engineering strategy with the in silico viscosity prediction screening step, we were able to efficiently improve the highly viscous anti-IL-13 candidate, successfully decreasing the viscosity at 150 mg/mL from 34 cP to 13 cP in a panel of 16 variants.

## 1. Introduction

While antibody-based therapeutics have a track record of successful advancement to patients [1,2], the protein sequence of the final drug product is rarely identical to that of the molecule that was discovered in the natural repertoire or display library. Protein engineering is often required to refine antibody sequences from discovery repertoires into developable therapeutics [3]. For example, amino acid residues that introduce chemical liabilities or instability to the molecule are identified and removed by making the necessary substitutions to eliminate a problematic property while retaining both binding to target and functional activity. For many liabilities, such as oxidation, isomerization, and deamidation [4], collective data have provided insights into which amino acid combinations will be problematic and which substitutions will mitigate the problem [5]. A similar knowledge base is being applied to engineering for colloidal stability, especially for viscosity of antibodies when formulated at a relatively high concentration, e.g., ≥150 mg/mL. Viscosity of therapeutic proteins is particularly relevant for high-dose therapeutics that are applied subcutaneously and often self-administered. In this application, it is either difficult to transfer viscous material through a syringe or uncomfortable for a larger volume of material at a lower concentration and viscosity to be administered [6,7]. Evidence in the literature points to protein surface properties as key contributors to colloidal instability, such as high viscosity, for example patches of uniformly charged or hydrophobic residues on the protein surface [8,9,10,11].

In addition to these physics-based approaches to viscosity prediction and remediation, there is the ongoing development of machine learning algorithms that predict viscosity from protein sequence after being trained on datasets linking diverse protein sequences to measured viscosity values [12,13,14,15,16,17,18,19]. We built a machine learning model for antibody viscosity prediction using an internal antibody dataset [20]. Our viscosity model is a classifier that predicts from protein sequences whether an antibody will have “HIGH” viscosity, defined as ≥15 centipoise (cP) at 150 mg/mL in our standard formulation buffer at pH 5.2. MMAb3 is a fully human monoclonal antibody targeting huIL-13 that was discovered through immunization of a transgenic mouse [21,22] and affinity-matured using the huTARG mammalian display platform [23]. The binding and functional profile of MMAb3 is excellent, with 34 fM binding to huIL-13 and cross-reactivity to IL-13 from a cynomolgus monkey. After MMAb3 was engineered to remediate hotspots (MMAb3.1), the model correctly predicted that it had a significant probability of HIGH viscosity: MMAb3.1 was recombinantly produced from CHO cells and concentrated to 150 mg/mL in our platform buffer (10 mM acetate, 9% sucrose, pH 5.2), and a viscosity of 33.8 cP was measured. To make sure the viscosity had not been increased while hotspots were being engineered, the viscosity of the parental MMAb3 molecule was also measured, only to reveal that it had a higher viscosity of 76.9 cP. Because antibodies that bound to this epitope were rare, instead of culling the molecule, we undertook an engineering campaign to reduce viscosity while maintaining function. We report successful therapeutic engineering through designs focused on the rational modulation of protein surface properties that were filtered through the predictive viscosity model to reduce the number of variants in the panel to a size that would be tractable for a high-concentration viscosity measurement.

## 2. Materials and Methods

### 2.1. Protein Analysis 

To identify potential causes of viscosity, a Schrodinger BioLuminate 2020-1 Protein Preparation [24,25,26] was performed on the PBD:7REW MMAb3 NHP IL-13 co-crystal structure, after the removal of the IL-13 antigen, with the pH set to 5.2. This was followed by running a Schrodinger BioLuminate 2020-1 protein surface analysis to identify and define surface patches of a hydrophobic, positively charged, or negatively charged nature [24,25,26]. Resulting surface patches were filtered for significant impact based on a size of >500 Å^2^ and a Schrodinger patch score of >500. In addition, residues were narrowed down to variants with a contribution of >40 kcal/mol (Appendix A). The Schrodinger contribution score is calculated as an energy score based on the collection of elements on the protein’s surface. Amino acid interactions in MMAb3 and between MMAb3 and IL-13 in PDB:7REW were analyzed and displayed using PyMol Schrodinger 2.2.0 software [27].

### 2.2. Viscosity Prediction 

Our viscosity prediction algorithm was trained on a set of internally developed mAbs [20]. The model development and evaluation strategy followed a process similar to that previously described [20] with the following modeling differences: (1) the model was trained using only our proprietary list of 206 hand-engineered features based on amino acid counts and calculated charge, aromatic content, and hydrophobicity in the various variable fraction (Fv) regions of the mAbs (CDRs, framework, etc.); (2) feature selection was performed by recursive feature elimination with random forest and a subset of 30 score-based features was selected in our final model; and (3) our deployed model was a random forest classifier. The model development workflow is summarized in Figure 1A. To generate predictions for anti-IL-13 MMAb3.1 mAb sequences [23], the workflow described in Figure 1B was followed. First, the tool takes as the input the Fv amino acid sequence. It then computes 30 sequence-based descriptors that represent three types of scores, charge, aromatic, and hydrophobic on various Fv regions. The tool then passes these as inputs to a trained random forest model to obtain predictions as either LOW or HIGH viscosity. 

### 2.3. Protein Production 

Candidate molecules were produced by DNA cloning into plasmids followed by recombinant mammalian expression and purification by Protein A capture. DNA plasmids were cloned using golden gate technology [28,29] to assemble antibody variable domains, purchased as G-block nucleotide strands, with immunoglobulin G (IgG) constant domains and insert them into proprietary recombinant expression vectors. In the HC containing plasmid, the selectable marker was the puromycin resistance gene, puromycin-N-acetyltransferase. In the LC containing plasmid, the selectable marker was the hygromycin resistance gene, hygromycin B phosphotransferase. Plasmids were transfected into a CHO-K1 cell line for expression using Lipofectamine LTX (Thermo Fisher Scientific, Waltham, MA, USA) by established lipid transfection methods with equal parts of mAb encoding plasmid to Piggybac transposase encoding plasmid (System Biosciences, Palo Alto, CA, USA). After recovery from selection by puromycin at 20 mg/mL and hygromycin at 500 mg/mL (Thermo Fisher Scientific, Waltham, MA, USA), cultures were scaled by dilution into a proprietary growth medium. Inoculation was performed by dilution into a proprietary front-loaded production medium at a density of 1.5 × 10^6^ cells/mL before production for 6 days. Clarified supernatants were affinity captured by MabSelect SuRe chromatography (Cytia, Piscataway, NJ, USA), using Dulbecco’s PBS without divalent cations (Thermo Fisher Scientific, Waltham, MA, USA) and 25 mM tris-HCl, 500 mM L-Arg-HCl, and pH 7.5 as the wash buffers, and 100 mM acetic acid at pH 3.6 as the elution buffer. All separations were carried out at ambient temperature. When the absorbance at 280 nm was above a defined threshold, the elution was immediately conditioned with 0.03 volumes of 2 M tris using an in-line mixer. Conditioning was terminated when the absorbance was below the defined threshold. The affinity pool was then filtered through a 0.22 μm cellulose acetate filter.

The pools were diafiltered against approximately 30 volumes of 10 mM sodium acetate, 9% sucrose, and pH 5.2 using Slide-A-Lyzer dialysis cassettes with a 20 kDa cutoff membrane (Thermo Scientific, Waltham, MA, USA) and further concentrated using a Centriprep centrifugal concentrator with a 30 kDa cutoff membrane (EMD Millipore, Burlington, MA, USA). The concentrated material was then filtered through a 0.8/0.2 μm cellulose acetate filter and the concentration was determined by the absorbance at 280 nm using the calculated extinction coefficient. Sample purity was determined by LabChip GXII analysis under reducing (with 32.7 mM) and non-reducing (with 25 mM of iodoacetamide) conditions. Analytical SEC was carried out using a BEH200 column (Waters, Milford, MA, USA) with an isocratic elution in 100 mM of sodium phosphate, 50 mM of NaCl, 7.5% ethanol, and pH 6.9 over 10’.

### 2.4. Viscosity Measurement 

Prior to cone and plate viscosity measurements, polysorbate 80 was added to a final concentration of 0.01%. Cone and plate viscosity measurements were performed at 25 °C on an Anton Paar Rheometer using standard methods [20]. 

### 2.5. Functional Assessment

Variants of anti-IL-13 MMAb3.1 were tested for retained function using a human PBMC IL-13 induced TARC assay. PBMCs were isolated from leukapheresis packs (HemaCare Corp, Northridge, CA, USA) by density centrifugation with Ficoll-Pague Plus^TM^ kits (Millipore Sigma Aldrich, St. Louis, MO, USA). PBMCs were resuspended in CPZ ^TM^ cryopreservation medium (Incell, San Antonio, TX, USA) at a cell density of 2 × 10^8^ cells/mL before being frozen down. To test the inhibition of IL-13 induced TARC by our MMAb3.1 variants, PMBCs were thawed and resuspended in assay media containing RPMI-1640, 10% heat inactivated FBS, and penicillin–streptomycin–glutamine. A total of 2 × 10^5^ cells/well were added to 96-well U-bottom tissue culture plates (Corning, Corning, NY, USA), then incubated for 2 h at 37 °C and 5% CO_2_ to recover. MMAb3.1 variants underwent a ten-point 1:3 serial dilution (100 nM to 0.005 nM) and equal volumes were pre-incubated with 30 ng/mL of human IL-13 (PeproTech, Cranbury, NJ, USA) for 20 min at room temperature. For data normalization and percent of control (POC) conversion, conditions lacking IL-13 and MMAb3.1 served as a low control and conditions lacking MMAb3.1 served as a high control. Resulting pre-incubates were added at a 1:5 ratio to 2 × 10^5^ PBMC and incubated for an additional 48 h at 37 °C, 5% CO2. A 50 mL supernatant was collected and added to TARC plates by V-PLEX Plus (Meso Scale Diagnostics, Rockville, MD, USA) and TARC was detected following the manufacturer’s instructions using a TARC SULFO-TAG detection antibody on a MESO SECTOR S 600 Meso Scale Diagnostics (MSD, Rockville, MD, USA) plate reader. To normalize data, the average of the baseline control values was subtracted from each MMAb3.1 dose–response curve value. The normalized dose–response values were converted to POC as follows:$$POC=\frac{\mathrm{normalized data value}}{\mathrm{average high control value}-\mathrm{average low control value}}\times 100$$

IC_50_ values were attained with GraphPad Prizm 8.4.3 software by determining the averages and standard deviations of duplicate samples and performing a non-linear curve fit analysis with a four-parameter variable slope.

## 3. Results 

To identify the residues contributing to the high viscosity of our anti-IL-13 MMAb3 mAb, a co-crystal structure of MMAb3 Fab in a complex with the NHP IL-13 target (PDB ID:7REW) was generated at a resolution of 2.1 angstrom [23]. The structure was used to calculate protein surface features and focus our amino acid substitutions on residues that were not directly involved in the epitope–paratope or structural interactions of Fab. To identify which residues to consider engineering, a protein surface analysis was performed on the MMAb3 Fab structure, without the IL-13 antigen, using the Schrodinger BioLuminate 2020-1 software package. The results of the surface analysis reveal an assortment of patches of various sizes and predicted impacts. The surface patches with a larger size in Å^2^ and higher “patch scores” were visually inspected on the surface of the structure for compactness of residues with significant impacts. From this inspection, two patches were selected for potential engineering (Appendix A). Of the two patches, the largest one, “Patch 44”, was a negatively charged patch positioned predominantly on the light chain (Figure 2A,B). The second largest patch, “Patch 9”, was positively charged and positioned predominantly on the heavy chain (Figure 2B,C). Amino acid residues that were key players in creating each surface patch, as measured by a calculated “contribution score” > 40 kcal/mol, were targeted for engineering (Appendix A). On Patch 44, this consisted of LC:Y2, LC:E3, LC:D26, LC:D33, LC:D67, LC:D87, and HC:E53. One residue in Patch 44 with a lower contribution score was also included (LC:D110), due to its potential to address an isomerization site in LC:CDR3. 

To focus our engineering on non-critical residues, interactive residues were identified in the MMAb3–IL-13 complex structure using the protein interaction analysis function in BioLuminate 2020-1 software. Interactions in the structure were then visually confirmed using PyMOL 2.2.0 software (Figure 3A). Of the residues in Patch 44 with high contribution scores, the LC:D67 residue was observed to interact directly with the IL-13 residue K74, playing a critical role in IL-13 antigen binding (Figure 3B). The interaction of LC:D67 with IL-13:K74 was also specified in the Schrodinger protein interaction analysis (data not included). The HC:E53 residue was observed to contribute to the HC structure by interacting with HC:R45 in the beta sheet (Figure 3C). Due to the crucial interactive roles of these residues, they were not modified. To identify potential alternate residues with minimal immunogenicity risk, the MMAb3.1 variable domain was aligned to related LC and HC germlines, VL3 and VH3, respectively, at http://antibody/docs/vbase.html (accessed on 4 June 2020). For the LC, alignment revealed the alternate sub-germline options of Y2S, E3V, D33S, D33K, D87N, and D87T. For the HC, most residues checked were highly conserved with the exception of HC:R97K.

A set of variants engineered for improved viscosity was narrowed down by entering their Fv sequences into our viscosity prediction software (Figure 1). Potential mutations to lower viscosity were evaluated individually and in combinations of two, three, and four mutations (Table 1 and Table 2). For positively charged Patch 9 with contributing residues HC:R20, HC:K86, and HC:R97, the results demonstrate that none of the individual or combinatorial mutations changed the prediction from a HIGH to LOW viscosity (Table 1). This demonstrated a low probability of engineering success, and as a result, corrections to the positively charged Patch 9 were not pursued any further. For negatively charged Patch 44, with contributing residues LC:Y2, LC:E3, LC:D26, LC:D33, LC:D87, and LC:D110, the results demonstrate that single and double mutants as well as many mutations to polar or germline residue predictions did not change from a HIGH to LOW viscosity (Table 2). Combinations of three and four mutations from negatively charged residues to positively charged residues frequently did result in a change in predictions from a HIGH to LOW viscosity, indicating a relatively high probability of engineering success (Table 2). In many combinatorial cases, mutations at LC:E3 and LC:D87 had the same prediction, whether substituted with the positively charged amino acid lysine or alternate sub-germline residues LC:E3V and LC:D87N, respectively. At these sites, only germline residues were advanced to minimize potential immunogenicity. Altogether, we were able to assess the probability of 74 candidate mutation combinations with our in silico prediction tool. From the viscosity prediction of mutations, we identified 11 variants along with 9 controls to advance for production followed by cone and plate viscosity measurements. 

Of the 20 variants that were advanced to production, 4 variants failed to express, leaving 16 variants for analysis. Cone and plate viscosity measurements were taken of the purified MMAb3.1 variants concentrated at 150 mg/mL: five variants had viscosity measured at <15 cP. The distribution of measured viscosity for these 16 MMAb3.1 variants is shown in Figure 4A. Furthermore, per variant measured viscosity versus prediction is shown in detail in Table 3. Table 4 summarizes the different performance metrics (the maximum value for accuracy is 100% and for all other metrics is 1; a larger value of these metrics indicates better performance). The accuracy of model predictions for the 16 tested variants was 81.25%. The predictive model achieved a precision of 1: all the engineered variants that were predicted to have HIGH viscosity had cone and plate measurements in the range of 19.8 to 33.8 cP at 150 mg/mL (Figure 4B, Table 3). A total of 11 of the 16 variants had a measured viscosity of ≥15 cP at 150 mg/mL. Of these, the model incorrectly predicted three to be LOW viscosity, hence achieving a recall of 0.73 (equal to the harmonic mean of precision; Table 4). The overall tradeoff between precision and recall is represented by the F1-score of 0.84.

Three engineered variants with low viscosity were tested for retained function in a human PBMC, IL-13-induced TARC (thymus and activation-regulated chemokine) (CCL17) assay, in which a serial dilution of candidate molecules had a dose–response inhibition of IL-13-induced TARC from PBMCs. The activity of the low-viscosity mutants, measured as IC_50_ of the TARC MSD signal, was within two-fold of the MMAb3 and MMAb3.1 controls of 35.85 and 39.28 pM, respectively, and ranged from 28.54 to 52.16 pM (Figure 5, Table 3). 

## 4. Discussion

Given the nature of the tool as a classifier, the key leverage gained from our viscosity prediction software was the ability to rationally perform a computational screen of many engineered variants, thereby decreasing the number of candidates with a high probability of low viscosity for a single large-scale round of production and biophysical characterization. This approach not only enabled us to obtain direct measures of viscosity for each candidate molecule; it also allowed us to generate enough material for a more complete characterization. As a result, the use of viscosity prediction increased our chance of success by providing a high throughput means to screen more variants and cull ones with a low potential for viscosity remediation. It also shortened the timeline for identifying the molecule with the best biophysical characteristics by bringing the number of candidates down to a level that could be screened and characterized in one round of protein expression, purification, and characterization rather than requiring a high throughput production and screen of many variants followed by a large-scale production and characterization of a few variants. In a platform consisting of stable cell line production, this would shorten the timeline by approximately two months. 

While viscosity prediction proved beneficial in this instance, a more broadly utility would be to predict the viscosity of newly obtained mAb sequences as soon as they are available. Potentially viscous mAbs can be sorted out from the low-viscosity mAbs and can then either be deprioritized or proactively engineered with similar cycles of in silico testing of engineered variants with the viscosity prediction algorithm. In addition, when molecules are engineered to mitigate other molecular liabilities, the resulting sequences can be screened for favorable viscosity before being advanced to laboratory production. The ability to screen for viscosity at the sequence level with high confidence before generating molecules is an empowering asset that can contribute to increased efficiency and success at engineering well-behaved antibodies and shortening product timelines.

## Figures and Tables

**Figure 1 biomolecules-14-00617-f001:**
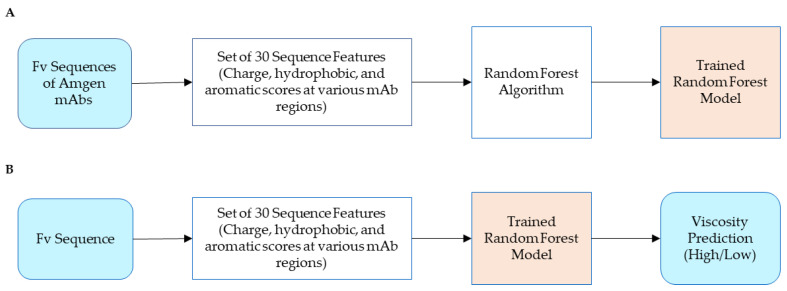
Workflow showing overview of the viscosity prediction tool. (**A**) Training of the internally developed viscosity prediction tool. The training set comprises the diverse sequences used for model building. (**B**) Steps involved in the viscosity prediction of any given sequence.

**Figure 2 biomolecules-14-00617-f002:**
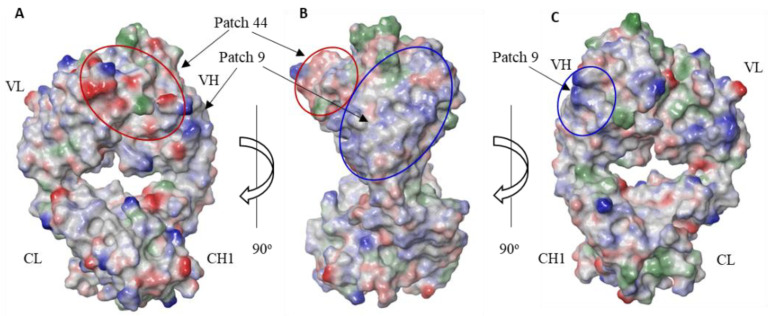
Surface patch analysis of MMAb3 Fab from PDB:7REW. (**A**) Surface view of MMAb3 with residues contributing to negatively charged Patch 44 colored and circled in red. (**B**) Surface view of MMAb3 after 90° rotation from (**A**) with residues contributing to Patch 44 colored and circled in red and residues contributing to positively charged Patch 9 colored and circled in blue. (**C**) MMAb3 after 90° rotation from (**B**) with contributing residues colored and circled in blue.

**Figure 3 biomolecules-14-00617-f003:**
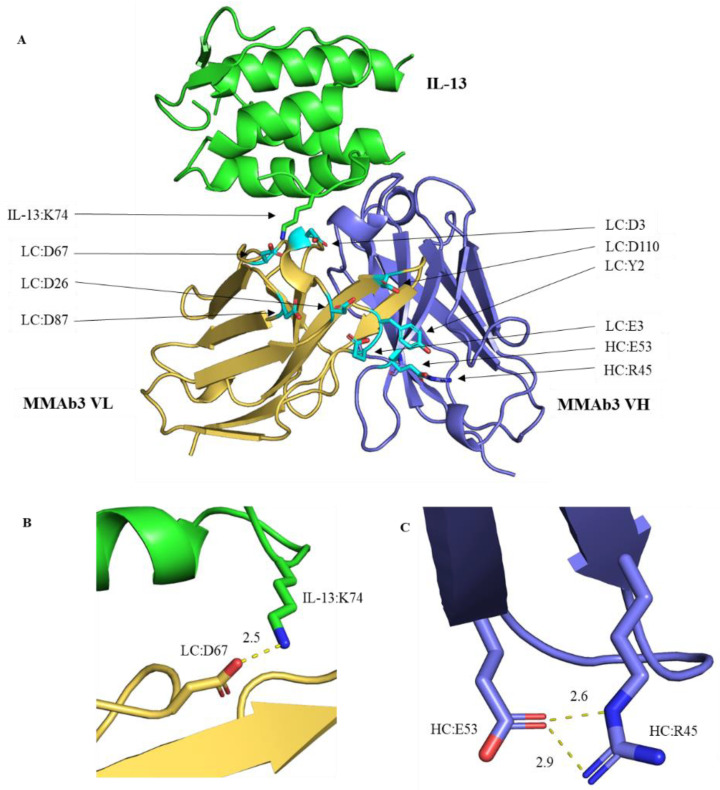
Pymol 2.2.0 co-crystal of MMAb3 Fab with NHP IL-13 from PDB:7REW. IL-13 antigen in green; MMAb3 LC in yellow-orange; MMAb3 HC in slate. (**A**) Residues with a significant contribution to negatively charged Patch 44 are colored in cyan, demonstrating the positioning of contributing residues across the LC of the Fab surface. (**B**) MMAb3 LC:D67 interacts with IL-13 K74 with complementary charges and a proximity of 2.5 Å. (**C**) MMAb3 HC:E53 interacts with HC:R45, contributing to the beta sheet structure with proximities of 2.6 Å and 2.9 Å.

**Figure 4 biomolecules-14-00617-f004:**
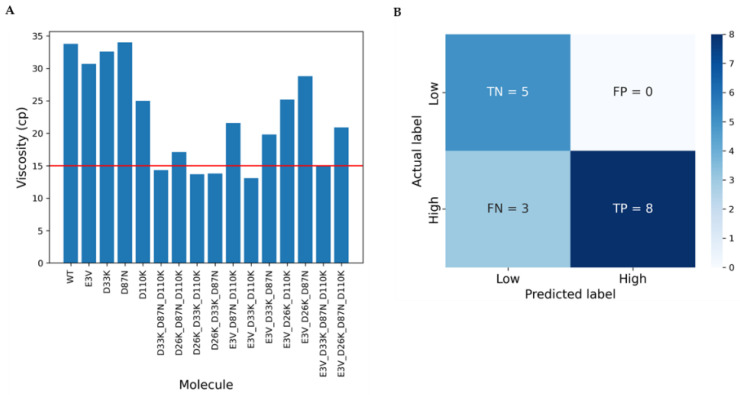
Predicted and measured viscosities for 16 MMAb3.1 variants. (**A**) The distribution of measured viscosities for the 16 MMAb3.1 variants. The red line denotes the 15 cP viscosity cutoff. Five of the variants had measured viscosities below the 15 cP cutoff. (**B**) Confusion matrix summarizing the performance on the IL13 dataset. TN = true negative (model predicted low and it is true), FP = false positive (model predicted high and it is false, i.e., actual measurement value is low), FN = false negative (model predicted low and it is false, i.e., actual measurement value is high), and TP = true positive (model predicted high and it is true).

**Figure 5 biomolecules-14-00617-f005:**
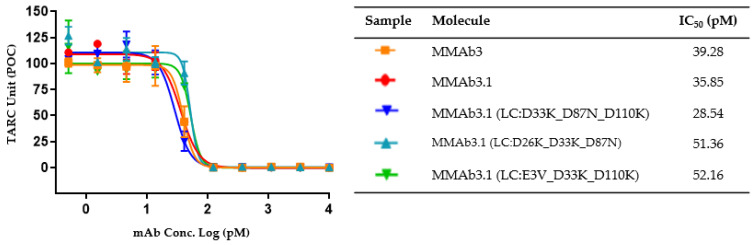
Functional assessment of lead candidates in the human PBMC, IL-13-induced TARC assay. Human IL-13 was pre-incubated with serially diluted mAb samples, then added to PBMCs and incubated for 48 h. TARC (CCL17) detection was measured from cell supernatants using the MSD platform with results normalized and graphed as percent of control (POC) for the inhibition of IL-13 induced TARC (CCL17). Results demonstrate an IC_50_ for viscosity engineered variants that is within range of the MMAb3 and MMAb3.1 parental clones.

**Table 1 biomolecules-14-00617-t001:** In silico viscosity prediction of MMAb3.1 positive-charge Patch #9 engineering variants. Hotspot positions in reference numbering and WT amino acid residues are listed in the header. Mutation amino acid residues and corresponding viscosity predictions are listed in the table.

HC:R20	HC:K86	HC:R97	Predicted Viscosity
			HIGH
S			HIGH
E			HIGH
	S		HIGH
		S	HIGH
		E	HIGH
S		S	HIGH
E		E	HIGH
S	S		HIGH
	S	S	HIGH
S	S	S	HIGH
	E		HIGH
E	E		HIGH
	E	E	HIGH
E	E	E	HIGH

**Table 2 biomolecules-14-00617-t002:** In silico viscosity prediction of MMAb3.1 negative-charge Patch # 44 engineered variants. Hotspot positions in reference numbering and WT amino acid residues are listed in header. Mutation amino acid residues and corresponding viscosity predictions are listed in the table. Variants that were generated in the lab for measurement are underlined.

LC:Y2	LC:E3	LC:D26	LC:D33	LC:D87	LC:D110	Predicted Viscosity
						HIGH
S						HIGH
	V					HIGH
		K				HIGH
			K			HIGH
				N		HIGH
					K	HIGH
S		K				HIGH
S			K			HIGH
S				N		HIGH
S				K		HIGH
S					K	HIGH
	V	K				HIGH
	V	N				HIGH
	V	S				HIGH
	V		K			HIGH
	V		S			HIGH
	V			N		HIGH
	V			K		HIGH
	V				K	HIGH
		K	K			HIGH
		N	K			HIGH
		K	S			HIGH
		K		N		HIGH
		K		K		HIGH
		N		N		HIGH
		K		T		HIGH
		K			K	HIGH
			K	N		HIGH
			K	K		HIGH
			S	N		HIGH
			K	T		HIGH
			K		K	HIGH
				N	K	HIGH
				K	K	HIGH
S		K	K			HIGH
S		K		N		HIGH
S		K			K	HIGH
S			K	N		HIGH
S			K		K	HIGH
S				N	K	HIGH
	V	K	K			LOW
	V	N	K			HIGH
	V	S	K			HIGH
	V	K	S			HIGH
	V	N	S			HIGH
	V	K		N		HIGH
	V	K		K		HIGH
	V	K			K	LOW
	V		K	N		HIGH
	V		K	K		HIGH
	V		K		K	LOW
	V			N	K	HIGH
	V			K	K	HIGH
		K	K	N		LOW
		N	K	N		HIGH
		K	S	N		HIGH
		K	K	T		LOW
		N	S	T		HIGH
		K	K		K	LOW
		K		N	K	LOW
		K		K	K	LOW
			K	N	K	LOW
			K	K	K	LOW
	V		K	N	K	LOW
	V		K	K	K	LOW
	V	K		N	K	LOW
	V	K		K	K	LOW
	V	K	K		K	LOW
	V	K	K	N		LOW
	V	K	K	K		LOW
		K	K	K	K	LOW
		K	K	N	K	LOW
		K	K	K	K	LOW

**Table 3 biomolecules-14-00617-t003:** Results of viscosity prediction, viscosity measure in cP, and functional activity in pM for lead variants. Concentrations at which viscosity measures were taken are listed in mg/mL.

LC:D26	LC:D33	LC:D87	LC:D110	Visc. Prediction	Conc. (mg/mL)	Visc. (cP)	TARC IC50 Ave. (pM)
				HIGH	150	33.8	39.28
				HIGH	150	30.7	
	K			HIGH	157	32.6	
		N		HIGH	150	34	
			K	HIGH	145	25	
	K	N	K	LOW	150	14.3	51.36
K		N	K	LOW	150	13.1	
K	K		K	LOW	150	13.7	
K	K	N		LOW	151	13.8	28.54
		N	K	HIGH	150	21.6	
	K		K	LOW	145	13.1	52.16
	K	N		HIGH	154	19.8	
K			K	LOW	150	25.2	
K		N		HIGH	153	28.8	
	K	N	K	LOW	150	14.9	
K		N	K	LOW	155	20.9	

**Table 4 biomolecules-14-00617-t004:** Different metrics quantifying model performance. Maximum value for accuracy is 100% and for all other metrics it is 1. Higher value of these metrics denotes a better performance.

Metrics	Formulae	Value
Accuracy	(TP + TN)/(TP + TN + FP + FN)	81.25%
Precision	TP/(TP + FP)	1
Recall	TP/(TP + FN)	0.73
F1-score	2 × (Precision × Recall)/(Precision + Recall)	0.84

## Data Availability

Supporting data are provided in the manuscript in a tabular form and in Appendix A.

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
