# Peer review of "Sequence-Based Viscosity Prediction for Rapid Antibody Engineering"

_biomolecules, 2024, doi:10.3390/biom14060617_

Round 1

Reviewer 1 Report

Comments and Suggestions for Authors

The introduction can be split into two paragraphs if possible.

Author Response

Hello Reviewer 1, 

Your constructive feedback on the “Sequence-based viscosity prediction for rapid antibody engineering” manuscript is greatly appreciated. The introduction section has been separated into two paragraphs with the first consisting of traditional protein engineering practices and expanding slightly to include the relevancy of viscosity of therapeutic proteins with two additional citations. The added content is in response to one of the reviewers. The second introduction paragraph consists of the building and implementation of a machine learning model and viscosity prediction tool. 

If any of your points were misunderstood or overlooked or if other potential improvements are recognized, we will be happy to polish the content further.

Thank you,

Bram

Reviewer 2 Report

Comments and Suggestions for Authors

Minor: in line 126: not sure why use upper case in "Polysorbate 80"

Author Response

Dear Reviewer 2, 

Your constructive feedback on the “Sequence-based viscosity prediction for rapid antibody engineering” manuscript is greatly appreciated. You noted the potential to correct Polysorbate 80 to all lower case. Thank you for bringing this to my attention. It has been revised on the current page 5, line 168. If any of your points were misunderstood or overlooked or if other potential improvements are recognized, we will be happy to polish the content further.

Best regards,

Bram

Reviewer 3 Report

Comments and Suggestions for Authors

This work follows MAbs. 2023; 15(1): 2256745.

Published online 2023 Sep 12. doi: 10.1080/19420862.2023.2256745 By Mock et al. Performed at Amgen.   This follow up paper established the always elusive connection between in silico and bench work. The work was sound, experiments are well conducted. Citations are correct.     The only comment I would offer is that in both the 2023 article and the present manuscript, especially, there seems to be a desire not just to predict but do it accurately in order to speed the process of antibody discovery. Nevertheless, "time" is not discussed. How is this faster? How much faster? How can this method improve the speed at which libraries are evaluated? How is viscosity associated with therapeutic outcomes? Not just the solubility and colloidal stability terms.   I also appreciate the effort made by the authors to explain appropriately the methods for others to replicate them with their own libraries.          

Author Response

Dear Reviewer 3, 

Your constructive feedback on the “Sequence-based viscosity prediction for rapid antibody engineering” manuscript is greatly appreciated. You noted potential to elaborate on the measured gains in speed from using viscosity prediction, additional utility of the prediction tool, and how viscosity is associated with therapeutic outcomes. To elaborate on the relevance of viscosity on therapeutic outcomes, sentences were added to the introduction on page 2, lines 58-62 as follows, “Viscosity of therapeutic proteins is particularly relevant for high dose therapeutics that are applied subcutaneously and often self-administered. In this application, it is either difficult to transfer viscous material through a syringe or uncomfortable for a larger volume of material at a lower concentration and viscosity to be administered.” References Bittner et al and Badkar et al have been added to support the relevance of viscosity and provide additional perspective. The version I see has the reference list coded in a way I can’t read. Is there something I need to do to add Bittner 2018 and Badkar 2021 to the reference list on the last page? To elaborate on the time saved with the use of the prediction tool in this specific case, sentences were added to the discussion on page 12, lines 324-332 as follows, “As a result, the use of viscosity prediction increased our chance of success by providing a high throughput means to screen more variants and cull ones with low potential for viscosity remediation. It also shortened the timeline for identifying the molecule with the best biophysical characteristics by bringing the number of candidates down to a level that could be screened and characterized in one round of protein expression, purification and characterization rather than requiring a high throughput production and screen of many variants followed by a large scale production and characterization of a few variants. In a platform consisting of stable cell line production, this would shorten the timeline by approximately two months.”  There is also potential to elaborate on further development of such tools with additional cycles of machine learning and feeding alternate hypothesis into the machine learning hypothesis. Whether that is within or beyond the scope of the manuscript can be considered.

If any of your points were misunderstood or overlooked or if other potential improvements are recognized, we will be happy to polish the content further.

Thank you,

Bram

Reviewer 4 Report

Comments and Suggestions for Authors

The manuscript reports theoretical prediction of antibody molecule viscosity based on a machine learning technique, which is an important progress in field of therapeutics as well as in basic bioengineering. The topics seems to fit this special issue and the results will interest many readers of this field. The adopted methods are sufficiently described. I would like to recommend this manuscript for publication.

I have just two minor comments, which can improve its readability:

(1) Line 150 on p. 4:

It would better to show the equation independently from the main text, like

              normalized data value

POC =  ------------------------------------------------------------------- X 100

        average high control value - average low control value

(2) Tables 1 and 2:

Brief description on symbols S, E V, K, N, T would be beneficial to readers in different fields (including me).

Author Response

Dear Reviewer 4,

Your constructive feedback on the “Sequence-based viscosity prediction for rapid antibody engineering” manuscript is greatly appreciated. Your first suggestion to separate the equation from the main text has been implemented on page 5, line 193. I think it looks much better that way. Your second note of the benefit for additional detail in amino acid symbols for quick comprehension by a diverse audience resonated quickly. It has been implemented with the following sentence added to  the abbreviations section on page 12, lines 353-355, “Amino acid residues are depicted by their single letter codes, with frequent use of D (aspartic acid), E (glutamic acid), R (arginine), K (lysine), S (serine), T (threonine), N (asparagine), V (valine) and Y (tyrosine).” In addition, abbreviations pertaining to the supplementary table were added to page 12, line 351 as follows, “FR, frame region; CDR, complementary determining region”. Content was added to the table legends including Table 1 on page 8, lines 267-269 with “Hotspot positions in reference numbering and WT amino acid residues are listed in the header. Mutation amino acid residues and corresponding viscosity predictions are listed within the table” and Table 2 on page 8, lines 270-273 with “Hotspot positions in reference numbering and WT amino acid residues are listed in the header. Mutation amino acid residues and corresponding viscosity predictions are listed within the table. Variants that were generated in the lab for measurement are underlined”. My hope is that these extra details will provide the building blocks for quick comprehension and the ability for people to comprehend from start to finish with few if any pauses. 

If any of your points were misunderstood or overlooked or if other potential improvements are recognized, we will be happy to polish the content further.

Thank you,

Bram